# Nuclear Magnetic Resonance Metabolomics Approach for the Analysis of Major Legume Sprouts Coupled to Chemometrics

**DOI:** 10.3390/molecules26030761

**Published:** 2021-02-02

**Authors:** Mohamed A. Farag, Mohamed G. Sharaf El-Din, Mohamed A. Selim, Asmaa I. Owis, Sameh F. Abouzid, Andrea Porzel, Ludger A. Wessjohann, Asmaa Otify

**Affiliations:** 1Pharmacognosy Department, Faculty of Pharmacy, Cairo University, Cairo 12613, Egypt; dr_nabilselim79@yahoo.com (M.A.S.); asmaa.otify@gmail.com (A.O.); 2Chemistry Department, School of Sciences & Engineering, The American University in Cairo, New Cairo 11835, Egypt; 3Pharmacognosy Department, Faculty of Pharmacy, Port Said University, Port Said 42526, Egypt; drmgm4824@yahoo.com; 4Pharmacognosy Department, Faculty of Pharmacy, Misr University for Science & Technology (MUST), 6th October City 12566, Egypt; 5Pharmacognosy Department, Faculty of Pharmacy, Beni-Suef University, Beni-Suef 62521, Egypt; asmaa_owis@yahoo.com (A.I.O.); sameh.zaid@pharm.bsu.edu.eg (S.F.A.); 6Department of Bioorganic Chemistry, Leibniz Institute of Plant Biochemistry, Weinberg 3, 06120 Halle (Saale), Germany; aporzel@ipb-halle.de

**Keywords:** metabolomics, NMR, fingerprint, quantification, legumes, sprouts, chickpea, fava, fenugreek, lentil

## Abstract

Legume sprouts are a fresh nutritive source of phytochemicals of increasing attention worldwide owing to their many health benefits. Nuclear magnetic resonance (NMR) was utilized for the metabolite fingerprinting of 4 major legume sprouts, belonging to family Fabaceae, to be exploited for quality control purposes. Thirty-two metabolites were identified belonging to different classes, i.e., fatty acids, sugars, amino acids, nucleobases, organic acids, sterols, alkaloids, and isoflavonoids. Quantitative NMR was employed for assessing the major identified metabolite levels and multivariate data analysis was utilized to assess metabolome heterogeneity among sprout samples. Isoflavones were detected exclusively in *Cicer* sprouts, whereas *Trigonella* was characterized by 4-hydroxyisoleucine. *Vicia* sprouts were distinguished from other legume sprouts by the presence of L-Dopa versus acetate abundance in *Lens*. A common alkaloid in all sprouts was trigonelline, detected at 8–25 µg/mg, suggesting its potential role in legume seeds’ germination. Trigonelline was found at highest levels in *Trigonella* sprouts. The aromatic NMR region data (δ 11.0–5.0 ppm) provided a better classification power than the full range (δ 11.0–0.0 ppm) as sprout variations mostly originated from secondary metabolites, which can serve as chemotaxonomic markers.

## 1. Introduction

The plants of family Fabaceae grow worldwide in different climatic regions and are considered to be the third largest family among the flowering plant families, with about 700 genera and 20,000 species. Legume seeds are widely incorporated in the human diet, especially in developing countries, for their rich nutrient content of protein, providing about 33% of human dietary protein, nitrogen, starch, dietary fiber, and minerals [1,2,3]. Legumes are enriched in phytochemicals, i.e., flavonoids, alkaloids, phenolic acids, and saponins, some of which have proven or proposed health-promoting action and medicinal importance, offering a protective effect against several chronic diseases, especially inflammation-based ones [4]. 

The presence of antinutrients in legume seeds, e.g., tannins, phytic acid, trypsin inhibitors, and hemagglutinins nevertheless limits the nutritional value of many legumes [5]. Also, alkaloids are considered antinutrients due to their potential harmful actions, e.g., vicine and convicine in *Vicia* cause acute hemolytic anemia (favism) in patients with G6PD deficiency [6]. Germination is a food processing technique, in addition to cooking and autoclaving, to improve the nutritional quality of legumes by decreasing the content of antinutritional factors. Compared to the raw seeds, sprouts have a higher digestibility, biological value, higher vitamins content, and better minerals bioavailability as a result of either the de novo biosynthesis or transformation of complex food materials into a more accessible form during the germination process, thus germination is considered a kind of pre-digestion [7,8,9]. Moreover, the essential amino acids content is found higher in sprouted legumes than suggested in the World Health Organization (WHO) standards (exceeds 40% of the total identified amino acids) [5]. They are also considered as a good source of ω-3 fatty acids and their intake is recommended for maintaining a desirable ω-6 to ω-3 ratio in the human body, as stated by the WHO and Food and Agriculture Organization (FAO) [7]. Aside from attenuating antinutrient levels, germination increases health-promoting phytochemical levels, e.g., chickpea isoflavones were found to be ca. 300 times higher than in chickpea seeds [10]. 

Sprouts can be produced in a short period through simple, inexpensive, and environmentally safe germination procedures, offering a healthy, high-protein, low-fat, and low-calories diet [11]. They are often consumed fresh in salads or as appetizers, giving a sweet tender taste and crunchy texture rather than fibrous old plants, and also could be included in low-cost food formulations [12].

Sprouts’ utilization originated in the Far East region and spread to Western Europe and America in the late 20th and 21st centuries, especially with the increasing awareness of seed sprout health benefits [13]. Growing interest is paid to the evaluation of the nutritional value and the biological importance of sprouts [14] as a result of increasing consumption of such highly nutritive food worldwide [15]. Epidemiological studies support a relationship between the consumption of food products rich in phyto-phenolics and a low incidence of coronary heart disease, atherosclerosis, cancer, and stroke [16,17,18,19]. Legumes and their sprouts are rich in alkaloids, flavonoids, and terpenoids/steroids, which have a myriad therapeutic effects such as hypoglycemic, antimicrobial, antioxidant, and anti-inflammatory activities [20,21,22].

Due to the widespread medicinal uses and nutritional importance of legume sprouts, the authentication and standardization of their extracts are very important to ensure their quality and purity. The seeds of different legumes can be distinguished from each other on the bases of morphological characters, while it is difficult in the case of sprouts and even more when in an extract. This increases the need for employing analytical techniques that can confirm its origin as well as its standardization for future incorporation in nutraceuticals. NMR (nuclear magnetic resonance) and MS (mass spectrometry) are widely utilized for metabolite identification and quantification. The selection of a particular technique is dependent on its intrinsic gains and drawbacks, e.g., NMR is preferred to MS as it is a non-destructive, highly selective technique that could be utilized for metabolite structure elucidation and absolute quantification with the least sample processing and loss [23]. NMR metabolomic analysis was used successfully in various functional food systems for quality control assessment, e.g., date fruits, *Nigella* seeds, and cinnamon bark [24,25,26]. ^1^H-NMR allows for the simultaneous detection of a wide range of primary and secondary metabolites, making it well suitable for metabolite profiling and standardization purposes. A direct comparison among the concentrations of all metabolites can be achieved in an NMR spectrum, as signal intensity is directly proportional to its compound molar concentration.

The aim of this work was to identify, quantify, and standardize the major metabolites in legume sprouts (chickpea, fava, fenugreek, and lentil) using the NMR approach. Finally, multivariate data analysis was employed to classify sprout samples according to the identified metabolites to be compared with the previously published LC-MS (liquid chromatography-mass spectrometry) and GC-MS (gas chromatography-mass spectrometry) classification models.

## 2. Results and Discussion

### 2.1. NMR Fingerprinting of Legume Sprouts

NMR-based metabolomics is a reliable technique widely employed for the identification and quantification of metabolite classes in crude food extracts [27], mostly supported by the development of 2D-NMR (two dimensional nuclear magnetic resonance) analysis that aids in structural elucidation. In the present study, the methanol extracts from *Cicer*, *Lens*, *Trigonella*, and *Vicia* sprouts were analyzed via ^1^H-NMR and assigned in accordance with literature data and in-house databases, and to some extent, further confirmed by 2D-NMR experiments, i.e., COSY (correlation spectroscopy), HSQC (heteronuclear single quantum coherence spectroscopy), and HMBC (heteronuclear multiple bond correlation) (Appendix A). The ^1^H-NMR spectra of the different extracts are shown in Figure 1, with their corresponding chemical shifts and signal multiplicities displayed in Table 1. A total of 32 metabolites were clearly detected with their corresponding structures, as shown in Figure 2.

In the ^1^H-NMR spectra of the different samples, most high-intensity signals expectedly belong to primary metabolites, i.e., fatty acids (**1**–**2**), sugars (**3**–**6**), amino acids (**7**–**19**), nucleobases (**20**), and organic acids (**21**–**23**). In contrast, much lower-intensity peaks can be assigned to secondary metabolites, especially in the aromatic region, including sterols (**24**), alkaloids (**25**), and isoflavonoids (**26**–**32**), as listed in Table 1, and shown in Figure 1.

The sugars anomeric region (δ 4.0–5.5 ppm) in the ^1^H-NMR spectra (Figure 1A and Appendix A) showed signals related to sucrose (δ 5.39), fructose (δ 4.03 and 4.10), α-glucose (δ 5.11), and β-glucose (δ 4.48) as the main sugars (compounds **3**–**6**) in sprouts [11]. Sucrose was highest in *Cicer* sprouts in agreement with GC-MS findings [28]. Fatty, organic, and amino acids were detected in the up-field region (δ 1.5–4.0 ppm) of the spectra (Figure 1A). Nearly all sprouts were found to be enriched in essential and non-essential amino acids. Aliphatic amino acids (**7**–**14**) included: alanine (δ 1.46 (H-3)), valine (δ 1.02 (H-4) and 1.06 (H-5)), threonine (δ 1.31 (H-4)), asparagine (δ 2.94 (H-2a) and 2.72 (H-2b)), and proline (δ 3.98 (H-1), 2.11 and 2.30 (H-2), 1.96 (H-3) and 3.25 and 3.37 (H-4)). In contrast, 4-hydroxy-isoleucine (δ 1.24 (H-5) and 0.99 (H-6)) was only detected in *Trigonella* extract (Figure 1A and Appendix A). Additionally, choline, a quaternary ammonium base (δ 3.21 (*N*-CH_3_)), and its oxidation product, betaine (δ 3.27 (*N*-CH_3_)), were detected in all sprouts (Figure 1A and Appendix A). Choline participates in lecithin formation involved in cell membrane stabilization and additionally in the biosynthesis of acetylcholine neurotransmitter [29]. Betaine reduces animal body fat and improves growth and food utilization [30]. Additionally, both compounds have antidiabetic actions rationalizing for the use of some sprouts in diabetes [31].

With regards to the down-field region (δ 6.0–9.5 ppm) (Figure 1B and Appendix A), 2 aromatic amino acids (**15** and **16**) were detected in all sprouts yet were not observed in *Cicer* samples. These amino acids were phenylalanine (δ 7.33 (H-3′/H-5′) and 7.28 (H-2′/H-6′)) and tyrosine (δ 6.76 (H-3′/H-5′) and 7.12 (H-2′/H-6′)). L-Dopa (**17**) (δ 6.73 (H-2′), 7.75 (H-5′) and 6.61 (H-6′)) was detected exclusively in *Vicia* sprouts [32], corroborating previous results using both LC-MS and GC-MS [28]. The essential amino acids tryptophan (**18**) (δ 7.19 (H-5), 7.33 (H-7), 7.10 (H-8), 7.05 (H-9) and 7.63 (H-10)) and histidine (**19**) (δ 7.75 (H-5) and 7.01 (H-6)) were detected in all examined sprouts [11]. The identified amino acids were previously listed in germinated chickpea and lentil seed amino acid profiles [9]. Histidine and tryptophan are precursors of histamine and serotonin respectively, and can play a role in the treatment of anxiety and depression [33,34].

According to previous investigations, the antihyperlipidemic and hypoglycemic properties of *Trigonella* seeds were strongly related to their amino acid composition, especially 4-hydroxyisoleucine (**10**), which was detected exclusively in *Trigonella* sprouts (Figure 1A and Appendix A) and is considered the precursor of sotolon (3-hydroxyl-4,5-dimethyl-2(5*H*)-furanone), the powerful aroma component in fenugreek [35,36]. Another potential hypoglycemic alkaloid detected in most sprouts was trigonelline (**25**), identified from δ 9.23 (H-2) and 4.44 (*N*-CH_3_) characteristic signals (Figure 1B and Appendix A) [37]. Trigonelline was detected in all legumes studied and in accordance with LC-MS findings [28]. Trigonelline is widely distributed in dry legume seeds [38], however, the previous studies concerning its presence in germinated legumes are scarce and only report on fenugreek [39]. In plants, trigonelline acts as a reserve molecule which turns into NAD (nicotinamide adenine dinucleotide) during the germination process of, e.g., coffee seeds [40], which is suggestive for a role in sprout germination in legumes. 

Several studies showed that sprouted seeds contain higher amino acid levels than their seeds, concurrent with other beneficial constituents, such as phenolic compounds. This would thus be reflected in improved antioxidant activity of the germinated seeds [41]. 

The ^1^H-NMR spectra showed low-intensity signals attributed to cytosine (**20**) (δ 5.70 (H-5) and 8.01 (H-6)) in all sprouts (Figure 1B and Appendix A). Nucleobases play an important part in the regulation of many physiological processes in the human body via the purine or pyrimidine receptors. Additionally, some cytosine derivatives were reported to possess diverse biological activities such as antimicrobial and anticancer properties [42,43].

The ^1^H-NMR spectral analysis also revealed distinct signals from unsaturated ω-3 and ω-6 fatty acids (**1** and **2**) assigned to the allylic CH_2_ (δ 2.05–2.09), *bis*-allylic CH_2_ (δ 2.77 and 2.81), and olefinic (δ 5.30–5.38) protons (Figure 1A and Appendix A). The complete primary metabolite profiles of legume sprouts were previously listed using GC-MS, with sucrose found more enriched in *Cicer* sprouts [28], and also confirmed herein by absolute NMR quantification (Table 2).

^1^H-NMR spectral analysis indicated that all examined legume sprouts encompassed common organic acids (**21**–**23**), including 4-aminobutyric acid (δ 2.36 (H-2), 1.88 (H-3), 2.96 (H-4)) and fumaric acid (δ 6.67 (H-2/H-3)), whereas acetic acid (δ 1.92 (CH_3_)) exhibited its highest abundance in *Lens* extract (Figure 1A and Appendix A) [44]. In accordance with recent studies, high levels of organic acids provide carbon building blocks for defensive compounds production in plant tissues and are likely to function as phytoalexins at the sprout stage critical in plant life [45]. Moreover, organic acids are important food components, responsible for organoleptic characteristics as well as food safety and quality determination [46].

Among the secondary metabolites detected in all sprouts, β-sitosterol (**24**) showed relatively weak up-field signals, identified from its characteristic methyl signals δ 0.72 (H-18), 1.02 (H-19), and 0.83 (H-26/H-27), in addition to the olefinic proton at δ 5.34 (H-6) (Figure 1A and Appendix A) [47]. 

One of the most remarkable classes of secondary metabolites in *Cicer* sprout and absent from other sprouts included isoflavones or phytoestrogens (**26**–**32**), showing distinct H-2 proton singlets in the down-field region (δ 8.0–8.3 ppm), with daidzein and genistein as major forms along with their methylated and malonyl-glycoside derivatives (Figure 1B and Appendix A), and in accordance with reported data [48]. All detected isoflavones showed a monosubstituted B-ring system revealed by signals at δ 6.99 (H-3′/H-5′) and 7.49 (H-2′/H-6′) ppm (Appendix A). Di-substituted ring-A structure signals for genistein-based isoflavones (**26**–**28**) were assigned based on a set of *meta*-coupled doublets (δ 6.23 (H-6) and 6.35 (H-8) ppm for biochanin-A (**26**), δ 6.52 (H-6) and 6.71 (H-8) ppm for genistin (**27**), and δ 6.50 (H-6) and 6.72 (H-8) ppm for malonyl-genistin (**28**)). In contrast, the appearance of two doublets and a doublet of doublet signals in isoflavones (**29**) (δ 6.86 (H-8), 8.05 (H-5) and 6.94 (H-6) ppm), (**30**) (δ 7.25 (H-8), 8.14 (H-5) and 7.19 (H-6) ppm), and (**31**) (δ 7.22 (H-8), 8.14 (H-5) and 7.27 (H-6) ppm) revealed a mono-substituted ring-A system and allowed to annotate these compounds as formononetin (**29**), daidzin (**30**), and malonyl daidzin (**31**), respectively (Appendix A). Other spectral information to distinguish between the two isoflavones was based on the ^13^C chemical shifts, with the C-ring carbonyl in genistein moieties appearing more down-field shifted (δ 183.7) than that of daidzein isoflavones (δ 179.3) (Table 1 and Appendix A) [49]. 

Regarding the annotation of malonyl-glucoside forms of isoflavones, assignment was based on the key malonyl CH_2_ signal (δ 3.17 ppm), concurrent with the down-field shifts of the H-6 and H-8 aromatic protons at δ 6.50 and 6.72 (malonyl-genistin (**28**)) and δ 7.22 and 7.27 (malonyl-daidzin (**31**)) respectively, with respect to their corresponding glucosides at δ 6.52 and 6.71 (genistin (**27**)) and δ 7.19 and 7.25 (daidzin (**30**)), respectively (Table 1 and Appendix A). These down-field shifts were attributed to the de-shielding effect of malonic acid attached to a hydroxy group of glucose and in agreement with reported data [49]. However, the unexpected up-field shift of malonyl-genistin H-6 (Appendix A) is attributed to the hydrogen bonding between the free carboxylate of malonic acid with the hydroxyl group at C-5 of genistin, resulting in a little shielding near H-6. This assumption is confirmed by observing that H-6 of malonyl-daidzin without a hydroxyl group at C-5 does not show this pattern (Appendix A) [49]. Malonyl isoflavone glucosides were previously detected in chickpea seeds [50].

Methylated isoflavones, i.e., biochanin-A (**26**) and formononetin (**29**) showed distinct methyl signals (δ 3.83) and were found to be the major forms among all isoflavones detected (Figure 1A), and in agreement with UPLC-MS (ultra-performance liquid chromatography-mass spectrometry) results [28]. Biochanin-A, formononetin, and their 7-*O*-glucosides were previously isolated from chickpea seeds and sprouts [51,52]. Increase in isoflavones level was observed upon sprouting, suggesting that their biosynthesis may be activated during the germination process [51].

Among other isoflavonoids detected in *Cicer*
^1^H-NMR spectrum was cicerin (**32**) (Figure 1A and Appendix A), characterized from a key signal for the dioxygenated methylene moiety at δ 5.98 (OCH_2_O), 2 *meta*-coupled aromatic protons at δ 5.98 (H-6) and 5.96 (H-8), and 2 signals at δ 6.37 (H-3′) and δ 6.80 (H-6′) [53]. Cicerin was proposed as an important phytoalexin that plays a significant role in *Cicer* defense against microbial infection, especially at sprout critical stage in the plant lifecycle [53].

Isoflavonoids are well-recognized for a myriad of biological effects, i.e., antioxidant, estrogenic, antimicrobial, antiosteoporosis, and anticancer properties [54], some of which are rare in other flavonoid subclasses, such as strong phytoestrogenic effects. Previous findings revealed that germination remarkably increased isoflavones content, when compared to raw seeds, and hence is likely to contribute to enhanced antioxidant or estrogenic effects. This suggests that the germinated *Cicer* seeds may be a promising functional food component being rich in isoflavonoids [50].

### 2.2. Quantification of Major Metabolites Detected Via ^1^H-NMR

^1^H-NMR was further used to determine the absolute amounts of the identified metabolites in legume sprout extracts for future standardization purposes. NMR has been utilized efficiently in many medicinal plants and food metabolites for quantification without standard requirements [23,26]. For each of the previously mentioned identified metabolites, the ability of ^1^H-NMR to recognize a single well-resolved signal further allowed for their unbiased absolute quantification in sprout samples (Appendix A). The concentrations of the identified metabolites were expressed as µg/mg dry powder in different legume sprout samples, as shown in Table 2.

Sugars represented the major metabolites in all sprouts with maximal levels observed in *Cicer* extract (468.3 μg/mg total sugars), and with sucrose amounting for the major sugar. The high sugar content adds to the palatable taste of *Cicer* sprout. The percentage of the identified sugars ranges from 38% to 47%, and in accordance with that previously stated for other sprouts [5,9,55].

Total choline and betaine levels were quantified in all specimens, reaching up to 119.1 μg/mg in *Vicia,* rationalizing for its use as a natural antidiabetic [31]. Similarly, total amino acids content reached its highest level in *Vicia* samples (266.5 μg/mg). The high amino acids content adds to the nutritional value of *Vicia* sprout. However, 4-hydroxyisoleucine (51.1 μg/mg) was detected exclusively in *Trigonella* sprout, which may be correlated to its potential antidiabetic effect.

The absolute quantification utilizing NMR also showed that the highest levels of trigonelline were detected in *Trigonella* and *Cicer* sprouts, amounting to ca. 25 and 18 µg/mg, respectively. *Trigonella* and *Cicer* sprouts were also the richest in ω-3 fatty acids, amounting to 21.7 and 20.1 µg linolenic acid equivalent/mg dry matter respectively, as shown in Table 2. All sprouts contained the desirable ω-6 to ω-3 ratio recommended by the WHO and FAO, agreeing with the previously stated ratio in *Trigonella* sprouts [7]. *Vicia* sprouts were found rich in the anti-parkinsonismic L-Dopa, amounting to 112 µg/mg, confirming previous reports in sprouts of *Vicia faba* varieties [32]. *Cicer* sprouts presented a good source of isoflavonoids (~350 µg/mg) with malonylated isoflavone glycosides, i.e., malonyl-daidzin and malonyl-genistin, amounting to 80.2 and 78.9 µg/mg of the dried sprout matter, respectively (Table 2).

To the best of our knowledge and compared to previous NMR studies, this study provides the first comprehensive NMR metabolites fingerprinting and standardization of 4 sprouted legumes for future quality control purposes.

### 2.3. ^1^H-NMR Data Multivariate Data Analyses

Multivariate analysis results point to an advantage of our comparative metabolomics approach to reveal sample relatedness. Principal component analysis (PCA) and orthogonal projection to latent structures-discriminant analysis (OPLS-DA) are often utilized to analyze large complex datasets in order to define the differences between groups of data or to interpret group differences in meaningful ways.

#### 2.3.1. Unsupervised Multivariate PCA of Full-Range ^1^H-NMR Data

PCA is an extensively used multivariate data analysis method for chemometrics. PCA was performed within the full ^1^H-NMR region (δ 11.0–0.0 ppm) (Figure 3) for all sprouts, with distance to the model (DModX) test used to verify the presence of outliers (Appendix A). PC1, representing the main principal component, accounted for 58% of the total variance. The PC1/PC2 scores plot (Figure 3A) revealed 3 major distinct clusters corresponding to the four examined sprouts. *Cicer* specimens were located on the far-right side of the plot (positive PC1), while the remaining samples were positioned on the left side (negative PC1). Discrimination of *Trigonella* specimens from *Vicia* and *Lens* was observed along PC2 (23% of the variance). The score plot showed good reproducibility for all sprout specimens, confirming a low technical variability for the extraction method. Metabolites accounting for specimen’s segregation in a PCA score plot were revealed from the loading plot (Figure 3B), displaying the most discriminatory ^1^H-NMR signals. Three major groups stood out in this plot. The first corresponded to the ^1^H-NMR signals for isoflavonoids (δ 6.96) and sucrose (δ 3.61 and 3.71), contributing positively to PC1, and were found more enriched in *Cicer*. The second showed a negative effect on PC1 from ^1^H-NMR signals, which were assigned to asparagine (δ 3.84) and 4-hydroxy-isoleucine (δ 1.24), negatively affecting PC2 and abundant in *Trigonella*. Sugars (δ 3.76 and 3.64) positively effect PC2 and were found abundant in all sprouts except *Cicer*, suggesting that such sugars may be glucose and/or fructose. Metabolites showing less influence according to the loading score emanated from ^1^H-NMR signals of L-Dopa (δ 6.73 and 6.75), which was found exclusively in *Vicia* sprouts and had a negative effect on PC1 and a positive one on PC2 (Figure 3B). To confirm that the discrimination between samples is mostly affected by such metabolites among sprouts, i.e., sucrose, 4-hydroxy-isoleucine, and asparagine, box plots were attempted for these metabolites using NMR detection (Appendix A). In agreement with the PCA results, the highest level of sucrose was found in *Cicer*, while *Trigonella* was the sprout most enriched in hydroxy-isoleucine and asparagine. Details on the absolute quantifications for all major compounds detected in all sprouts are provided in Table 2. PCA results were further confirmed by performing a heatmap plot, which revealed a similar clustering pattern (Appendix A).

#### 2.3.2. Unsupervised Multivariate PCA of the Aromatic ^1^H-NMR Region Data

For more sample classifications and metabolite marker determinations, PCA was performed for all samples limited to the more distinctive aromatic ^1^H-NMR region (δ 11.0–5.0 ppm). Such model (Appendix A) showed better classification power than that of the full-range-based one with higher PC1 value (61%). As observed in full-range NMR, three distinct clusters were revealed in the PC1/PC2 scores plot (Appendix A), with *Cicer* specimens still being the most distant and located on the far-left side of the plot (negative PC1 values), whereas other sprouts were positioned at the right side (positive PC1). *Vicia* samples could be discriminated from *Trigonella* and *Lens* along PC2 (30% of total variance). The observed separation could be explained from the corresponding loading plot (Appendix A). In detail, high isoflavonoids (δ 6.99 and 7.49) content was detected in *Cicer* specimens contributing negatively to PC1, whereas L-Dopa (δ 6.61, 6.72, and 6.75) affected PC2 positively and was abundant in *Vicia* samples *(*Appendix A). Tryptophan (δ 7.33) showed less influential loading scores with positive effect on PC1, discriminating *Lens* and *Trigonella* sprouts (Appendix A). A similar clustering pattern was revealed from a heatmap plot (Appendix A), and in accordance with PCA results. Moreover, box plots’ results for isoflavonoids, l-Dopa, and tryptophan (Appendix A), as the major discriminatory metabolites, were in agreement with the PCA results.

#### 2.3.3. Supervised Multivariate OPLS-DA of ^1^H-NMR Data

In spite of the clear separation observed in both full and aromatic ^1^H-NMR-based PCA, legume metabolite markers were further confirmed by constructing several supervised OPLS-DA models. OPLS-DA is more potent in the identification of markers by providing the most relevant variables for the differentiation between two sample groups. First, *Cicer* samples were modelled against other sprout samples and analyzed using both ^1^H-NMR full region (δ 11.0–0.0 ppm) and aromatic region (δ 11.0–5.0 ppm) (Figure 4 and Appendix A, respectively). The derived score plot showed a clear separation of *Cicer* from other samples, with variance coverage of R^2^ = 0.95 (full range) and 0.97 (aromatic range), and a prediction goodness parameter of Q^2^ = 0.94 (full range) and 0.97 (aromatic range) (Figure 4A and Appendix A). The corresponding derived S-plot (Figure 4B and Appendix A), showing the contributing ^1^H-NMR signals, revealed that *Cicer* was particularly rich in sucrose (δ 3.61 and 3.71) and isoflavonoids (δ 6.99, 7.49, 7.84, and 8.12–8.16), where axes plotted from the predictive component are the covariance p[1] against the correlation p(cor)[1].

The PCA and OPLS-DA clustering of ^1^H-NMR data of legume sprouts confirmed the unique metabolite profile of *Cicer* in both primary and secondary metabolites which had previously already appeared in UPLC-MS and GC-MS data analyses [28]. The results suggested *Cicer* sprouts as a good source of estrogenic isoflavones [52].

To confirm the metabolic marker of *Trigonella*, appearing on the far-left side of the PCA plot (Figure 3A), *Trigonella* sprout was modelled against the other sprout samples and analyzed using both ^1^H-NMR full-region data (δ 11.0–0.0 ppm) and aromatic-region data (δ 11.0–5.0 ppm) (Figure 5 and Appendix A, respectively). The derived score plots revealed a clear discrimination between *Trigonella* and the remaining sprouts (Figure 5A and Appendix A). The corresponding S-plots (Figure 5B and Appendix A) showed that 4-hydroxy-isoleucine (δ 1.24), asparagine (δ 3.84), and trigonelline (δ 9.23 and 8.88) were abundant in *Trigonella*. The study confirmed that *Trigonella* sprouts exclusively contain 4-hydroxy-isoleucine in addition to being the richest in trigonelline alkaloid, both are suggested to mediate for the potential anti-diabetic and antihyperlipidemic actions of *Trigonella* sprouts [56,57]. This is in accordance with our previous UPLC-MS and GC-MS analyses [28] and further confirms ^1^H-NMR for absolute quantification (Table 2).

*Vicia* and *Lens* full-region ^1^H-NMR data (δ 11.0–0.0 ppm) and aromatic-region data (δ 11.0–5.0 ppm) were modelled against each other using OPLS-DA with derived score plots (R^2^ = 0.99 and Q^2^ = 0.99), showing a clear separation between both sample groups (Figure 6A and Appendix A). The corresponding derived S-plot (Figure 6B and Appendix A) showed that *Vicia* was particularly rich in sugars (δ 3.44–3.76 and 4.00–4.04), which, however, may depend on growing conditions, in addition to the exclusive and more specific presence of L-Dopa (δ 3.73, 3.75, and 6.61), whereas *Lens* was higher in acetate (δ 1.92), in agreement with ^1^H-NMR absolute quantification (Table 2). The study succeeded in distinguishing between *Vicia* and *Lens* samples and identification of each sample marker, along with confirmation of the exclusive presence of the anti-parkinsonism agent, L-Dopa, in *Vicia* samples, as previously revealed in UPLC-MS and GC-MS analyses [28].

## 3. Materials and Methods

### 3.1. Plant Material

Legume seeds: chickpea (*Cicer arietinum* L. cv. Giza 88), fenugreek (*Trigonella foenum-greacum* L. cv. Giza 2), fava (*Vicia faba* L. cv. Giza 3), and lentil *(Lens esculenta* L. cv. Sinai 1), were obtained from The Food Legumes Research Department, Field Crops Research Institute (FCRI), Agricultural Research Center (ARC), Giza, Egypt, in May 2014. The plants were cultivated in winter (early November 2013) and harvested fully ripe and dry in spring (late April 2014). Voucher specimens are kept at the Pharmacognosy Department Herbarium, Cairo University, Egypt.

### 3.2. Sprouting Procedures

The sprouting process was performed following the procedure described in Lv et al. [51]. In brief, 100 g of the dried seeds were soaked in 3 volumes of distilled water in glass containers for 8 h at 28 °C, followed by sprouting in glass dishes lined with cotton in the dark. The seeds were moistened with distilled water every 3 h during the germination process and washed twice daily for 3 days to avoid microbial growth. The seedlings were pinched, lyophilized, and then kept at −20 °C until further analysis. Sprouting was carried out in 3 independent biological replicates.

### 3.3. Chemicals and Reagents

Methanol-*d*_4_ (99.80% D) and hexamethyldisiloxane (HMDS) were purchased from Deutero GmbH (Kastellaun, Germany).

### 3.4. Extraction Procedure and Sample Preparation for NMR Analysis

A one-pot extraction protocol developed by Farag et al. [23] was employed for legume sprout extraction. The lyophilized and deep-frozen legume sprouts were ground with a pestle in a mortar under liquid nitrogen. The powder (120 mg) was homogenized with 5 mL 100% methanol using a Turrax mixer (11,000 RPM) 5 times for 20 s, with 1 min intervals to prevent heating. Extracts were then intensely vortexed and centrifuged at 3000× *g* for 30 min to remove sprout debris. 3 mL were aliquoted, and the solvent was evaporated under nitrogen until complete dryness. Dried extracts were resuspended with 800 μL 100% methanol-*d*_4_ containing HMDS (0.94 mM final concentration), and then centrifuged (13,000× *g* for 1 min). The supernatant was transferred to a 5 mm NMR tube. 3 biological replicates were analyzed under identical conditions for each specimen.

### 3.5. NMR Analysis

All spectra were recorded on an Agilent VNMRS 600 NMR spectrometer using a 5 mm inverse detection cryoprobe, and with the following parameters: frequency 599.83 MHz, digital resolution 0.367 Hz/point, pulse width 3 μs (45°), acquisition time 2.7 s, relaxation delay 23.7 s, number of transients 160, zero filling up to 128 K, and exponential window function with lb 0.4. 2D-NMR spectra were recorded using standard CHEMPACK 4.1 pulse sequences (gDQCOSY, gHSQCAD, gHMBCAD) implemented in Varian VNMRJ 2.2C spectrometer software. The heteronuclear single quantum coherence spectroscopy (HSQC) experiment was optimized for ^1^J_CH_ = 146 Hz with DEPT (distortionless enhancement by polarization transfer)-like editing and ^13^C-decoupling. The heteronuclear multiple bond correlation (HMBC) experiment was optimized for a long-range coupling of 8 Hz, and a two-step ^1^J_CH_ filter was used (130–165 Hz). Samples were randomly allocated in the sequence run.

### 3.6. NMR Quantification

For metabolite quantification using NMR spectroscopy, the peak areas of the internal standard (HMDS) and selected proton signals belonging to the target compounds were integrated manually for all samples. The following equation was applied for calculating metabolite concentrations (µg/mg dry matter):$$m_{T}=M_{T}\times \frac{I_{T}}{I_{\mathrm{St}}}\times \frac{X_{\mathrm{St}}}{X_{T}}\times C_{\mathrm{St}}\times V_{\mathrm{St}}$$

m_T_: mass of the target compound (µg) in the solution used for ^1^H-NMR measurement,

M_T_: molecular weight of the target compound (g/mol),

I_T_: relative integral value of the ^1^H-NMR signal of the target compound,

I_St_: relative integral value of the ^1^H-NMR signal of the standard compound,

X_St_: number of protons belonging to the ^1^H-NMR signal of the standard compound,

X_T_: number of protons belonging to the ^1^H-NMR signal of the target compound,

C_St_: concentration of internal standard (HMDS) in the solution used for ^1^H-NMR measurement (mmol/L),

V_St_: volume of solution used for ^1^H-NMR measurement (mL).

Signals used for NMR quantification are listed in Appendix A.

### 3.7. NMR Data Processing and Multivariate Data Analysis

The methodology used in this study was applied following the protocol of Farag et al. [23]. Briefly, the ^1^H-NMR spectra were automatically Fourier-transformed to (.esp) files using ACD/NMR Manager lab version 10.0 software (Toronto, ON, Canada). Spectral intensities were reduced to integrated regions (buckets) of equal width, 0.04 ppm, within the region of δ = 11.4−0.4 ppm. PCA was performed with R package (2.9.2) using custom-written procedures after scaling to HMDS signal, as described elsewhere [58]. OPLS-DA was performed with the program SIMCA-P Version 13.0 (Umetrics, Umeå, Sweden). All variables were mean-centered and scaled to Pareto variance. To assess the validity of the NMR-based OPLS models, Q^2^ and R^2^ values of all calculated models were bigger than 0.4 and close to 1, with most models showing a regression line crossing zero, with negative Q^2^ and R^2^ close to 1, which signifies the model’s validation. Also, the *p*-values for each OPLS-DA model were calculated using CV-ANOVA (analysis of variance of cross-validated residuals) and were all below *p*-value of 0.005 (Appendix A).

### 3.8. Statistical Analysis

NMR quantification data were analyzed using the Co-Stat computer program (version 8, Monterey, CA, USA). Data are expressed as mean ± standard deviation (SD) of the groups. Differences between sample groups were compared by one-way analysis of variance (ANOVA) and were considered statistically significant when *p* ≤ 0.05.

## 4. Conclusions

This research provided the first NMR-based metabolite fingerprinting of 4 major legume sprouts, i.e., *Cicer*, *Lens*, *Trigonella*, and *Vicia*. A total of 32 compounds belonging to various metabolite classes were identified and quantified. PCA and OPLS-DA were used for exploring the variations and determining the main markers of each sprout to be utilized in samples’ authentication and future quality control. Trigonelline and 4-hydroxy-isoleucine were found more enriched in *Trigonella* versus higher isoflavonoids and sucrose abundance in *Cicer* sprout. Nevertheless, sucrose cannot be considered as a useful marker as it is both a primary metabolite and quantitatively strongly dependent on growth conditions. *Vicia* was characterized by the exclusive presence of L-Dopa versus acetate abundance in *Lens*. The aromatic region data (δ 11.0–5.0 ppm) provided a better classification model than the full-range NMR (δ 11.0–0.0 ppm) as legume sprout variations mainly originated from secondary metabolites, which can serve as chemotaxonomic markers.

Determination of the metabolite patterns at different sprouting stages should now follow to provide a better understanding of the role of these constituents in the sprouting process, and to identify the optimum time of harvest for a certain effect or metabolite enrichment level. Moreover, future work should now examine different varieties or seed origin for each legume seed to determine whether differences in sprout composition shall be observed and/or to identify accessions yielding highest targeted metabolite levels.

## Figures and Tables

**Figure 1 molecules-26-00761-f001:**
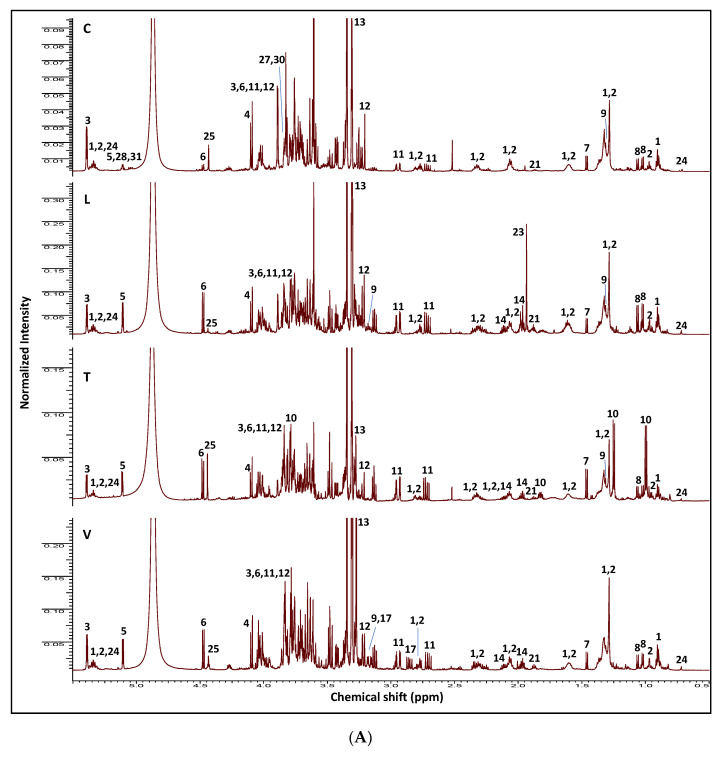
(**A**) ^1^H-NMR spectrum of legume sprout methanol extracts (C, *Cicer*; L, *Lens*; T, *Trigonella*; V, *Vicia*) showing characteristic signals for primary and secondary metabolites in the range δ 0.5–5.5 ppm, and (**B**) in the range δ 5.6–9.3 ppm. The identities of NMR peaks are listed in Table 1.

**Figure 2 molecules-26-00761-f002:**
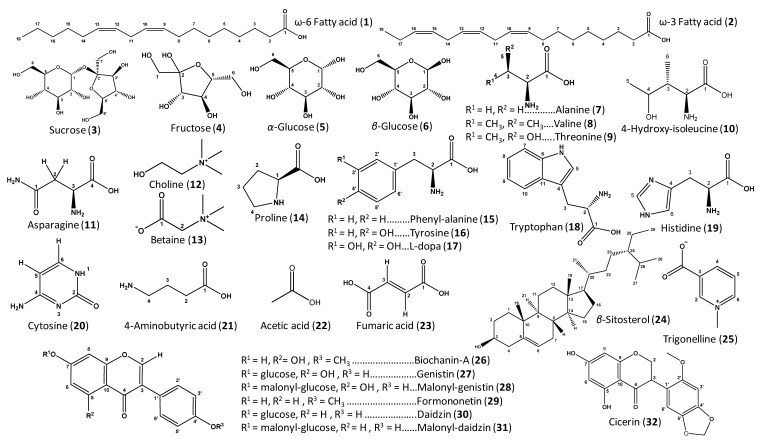
Structure of the major primary and secondary metabolites detected in chickpea, lentil, fenugreek, and fava sprout methanol extracts. Carbon numbering system for each compound is based on analogy rather than IUPAC rules. Metabolite numbers follow those listed in Table 1 for metabolite identification using 1D- and 2D-NMR.

**Figure 3 molecules-26-00761-f003:**
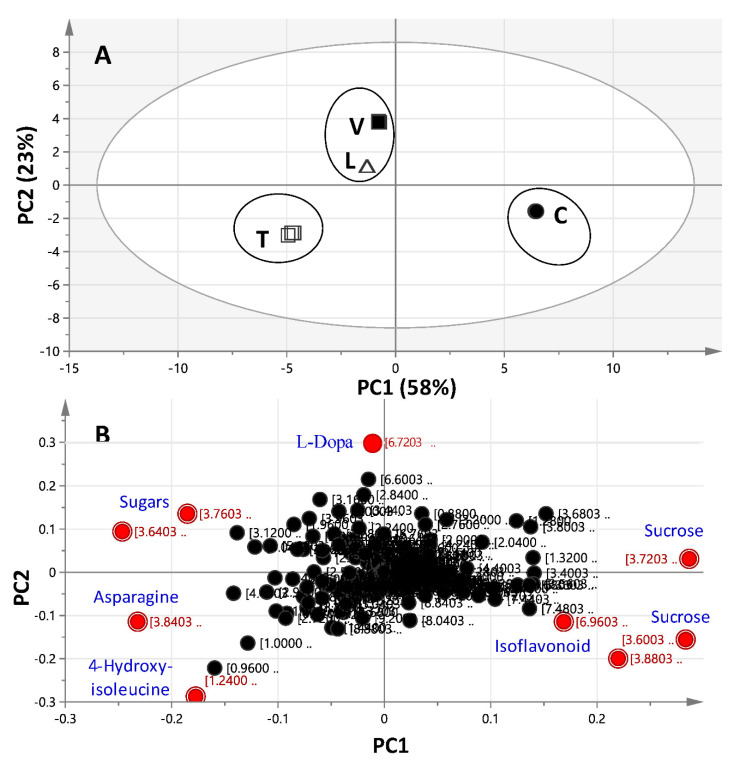
^1^H-NMR (δ 11.0–0.0 ppm)-based principal component analysis (PCA) of the four studied legume sprouts: *Cicer* (C), *Lens* (L), *Trigonella* (T), and *Vicia* (V) (*n* = 3). The clusters are located at distinct positions in two-dimensional space described by two vectors of principal components PC1 (0.58) and PC2 (0.23). (**A**) Score plot of PC1 vs. PC2 scores. (**B**) Loading plot for PC1 and PC2 contributing ^1^H-NMR signals and their assignments, with each metabolite denoted by its chemical shift (ppm). It should be noted that ellipses do not denote statistical significance but are rather added for better visibility of clusters discussed. For sample codes, refer to Table 1.

**Figure 4 molecules-26-00761-f004:**
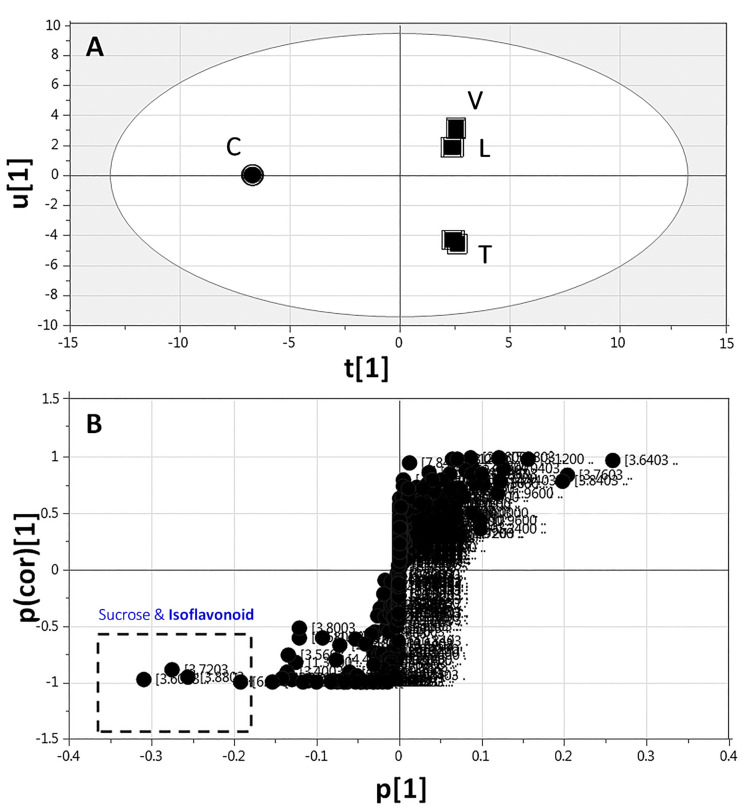
Full-range ^1^H-NMR (δ 11.0–0.0 ppm)-based orthogonal projection to latent structures-discriminant analysis (OPLS-DA) of *Cicer* sprouts (●) modelled against the remaining legume sprouts (■) (*n* = 3). (**A**) OPLS-DA score plot and (**B**) loading plot derived from samples modelled against each other. The loading S-plot shows the covariance p[1] against the correlation p(cor)[1] of the variables of the discriminating component of the OPLS-DA model. Peak numbering follows those listed in Table 1 for metabolite identification using 1D- and 2D-NMR.

**Figure 5 molecules-26-00761-f005:**
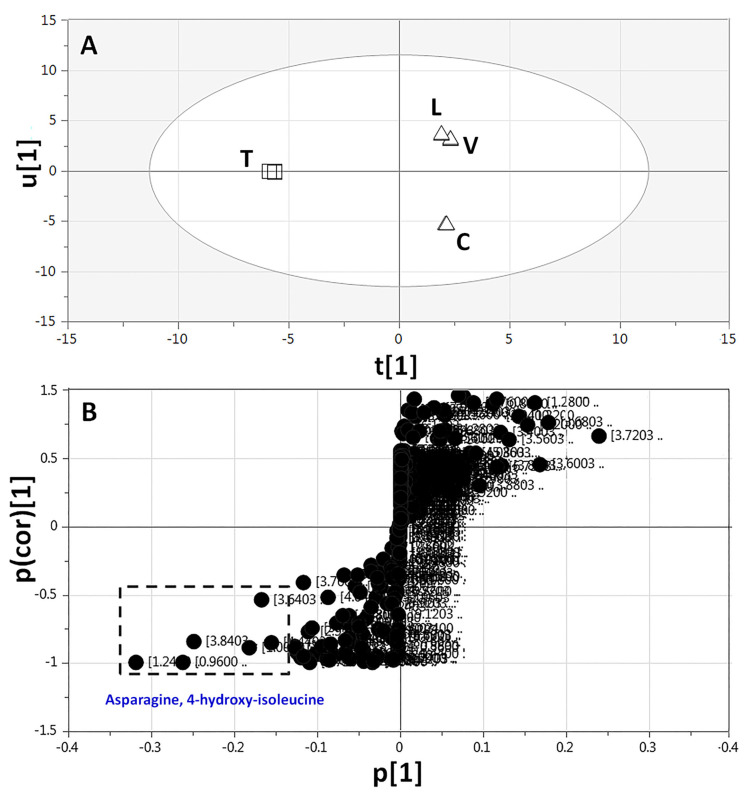
Orthogonal projection to latent structures-discriminant analysis (OPLS-DA) based on full-range ^1^H-NMR (δ 11.0–0.0 ppm) of *Trigonella* sprouts (□) modelled against the remaining legume sprouts (∆) (*n* = 3). (**A**) OPLS-DA score plot and (**B**) loading plot derived from samples modelled against each other. The loading S-plot shows the covariance p[1] against the correlation p(cor)[1] of the variables of the discriminating component of the OPLS-DA model. Peak numbering follows those listed in Table 1 for metabolite identification using 1D- and 2D-NMR.

**Figure 6 molecules-26-00761-f006:**
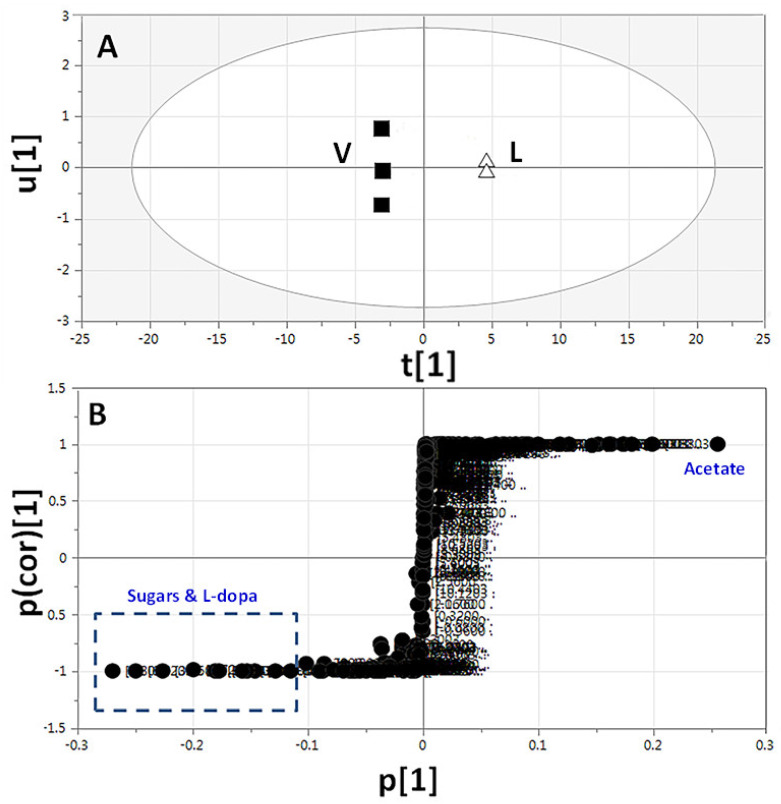
Orthogonal projection to latent structures-discriminant analysis (OPLS-DA) based on full-range ^1^H-NMR (δ 11.0–0.0 ppm) of *Vicia* sprouts (■) modelled against *Lens* sprouts (∆) (*n* = 3). (**A**) OPLS-DA score plot and (**B**) loading plot derived from samples modelled against each other. The loading S-plot shows the covariance p[1] against the correlation p(cor)[1] of the variables of the discriminating component of the OPLS-DA model. Peak numbering follows those listed in Table 1 for metabolites identification using 1D- and 2D-NMR.

**Table 1 molecules-26-00761-t001:** Resonance assignments with chemical shifts of constituents identified in 600 MHz ^1^H-NMR, HSQC, COSY, and HMBC spectra of legume sprout methanol extracts.

ID	Metabolite	Assignment	δ^1^H (ppm)	δ^13^C in HSQC (ppm)	δ^1^H in COSY (ppm)	HMBC correlations δ^13^C (ppm)	C	L	T	V
**1–2**	**ω-6 and ω-3 Fatty acids**	(CH_2_)n	1.27–1.39 (br. s)	30.8	0.91 (t-CH_3_), 1.61 (H-3), 2.07 (allylic CH_2_)	30.8 (CH_2_)n., 131.9 (olefinic C)	+	+	+	+
		C-2	2.31 (m)	35.5	1.61 (H-3)	26.2 (C-3), 30.8 (CH_2_)n, 175.8 (C-1)				
		C-3	1.61 (m)	26.2	2.31 (H-2), 1.33 (CH_2_)n	30.8 (CH_2_)n, 35.5 (C-2), 175.8 (C-1)				
		Olefinic Cs	5.30–5.38 (br. m)	129–132	2.77, 2.81 (*bis*-allylic CH_2_), 2.07 (allylic CH_2_)	26.8 (*bis*-allylic CH_2_), 130.2 (olefinic C)				
		allylic CH_2_	2.05–2.09 (m)	28.2–28.8	1.37 (CH_2_)n, 5.30–5.38 (olefinic Hs)	14.9 (t-CH_3_), 30.8 (CH_2_)n, 130.2, 131.9 (olefinic Cs)				
**1**	**ω-6 Fatty acid**	*bis*-allylic CH_2_	2.77 (t, *J* = 6.6 Hz)	26.8	5.30–5.38 (olefinic Hs)	130.2, 131.9 (olefinic Cs)				
		t-CH_3_	0.91 (t, *J* = 6.2 Hz)	14.9	1.33 (CH_2_)n	23.9 (ω-2 C)				
**2**	**ω-3 Fatty acid**	*bis*-allylic CH_2_	2.81 (t, *J* = 6.9 Hz)	26.8	5.30–5.38 (olefinic Hs)	130.2, 131.9 (olefinic Cs)				
		t-CH_3_	0.97 (t, *J* = 7.5 Hz)	14.9	2.07 (allylic CH_2_)	22.1 (ω-2 C), 133.7 (olefinic C)				
**3**	**Sucrose**	C-1	5.39 (d, *J* = 3.8 Hz)	94.5	3.42 (H-2)	73.9 (C-2), 105.6 (C-2′)	+	+	+	+
		C-2	3.42 (dd, *J* = 9.8, 3.8 Hz)	73.9	5.39 (H-1), 3.71 (H-3)	75.3 (C-3)				
		C-1′	3.61 (s)	64.7	-	79.9 (C-3′), 105.6 (C-2′)				
**4**	**Fructose**	C-3	4.10	78.6	4.03 (H-3)	62.0 (C-1), 74.6 (C-4)	+	+	+	+
		C-4	4.03 *	74.6	4.10 (H-4)	-				
**5**	**α-Glucose**	C-1	5.11 (d, *J* = 3.7 Hz)	94.5	3.35 (H-2)	73.2 (C-5), 74.7 (C-2)	+	+	+	+
		C-2	3.35 (br. s)	74.7	5.11 (H-1), 3.67 (H-3)	-				
**6**	**β-Glucose**	C-1	4.48 (d, *J* = 7.8 Hz)	97.9	3.13 (H-2)	75.2 (C-2), 77.3 (C-3)	+	+	+	+
		C-2	3.13 (dd, *J* = 7.8, 9.0 Hz)	75.2	3.35 (H-3)	77.3 (C-3)				
**7**	**Alanine**	C-2	3.59 *	51.9	1.46	17.6 (C-3), 176.6 (C-1)	+	+	+	+	
		C-3	1.46 (d, *J* = 7.2 Hz)	17.6	3.59	51.9 (C-2), 176.6 (C-1)					
**8**	**Valine**	C-2	3.42 *	61.9	2.25 (H-3)	19.6 (C-5), 30.7 (C-3)	+	+	+	+	
		C-3	2.25 (m)	30.7	1.02 (H-4), 1.06 (H-5), 3.42 (H-2)	180.2 (C-1)					
		C-4	1.02 (d, *J* = 7.0 Hz)	18.2	2.25 (H-3)	19.6 (C-5), 30.7 (C-3), 61.9 (C-2)					
		C-5	1.06 (d, *J* = 7.0 Hz)	19.6	2.25 (H-3)	18.2 (C-4), 30.7 (C-3), 61.9 (C-2)					
**9**	**Threonine**	C-2	3.18 *	nd	4.12 (H-3)	-	+	+	+	+	
		C-3	4.12 *	nd	1.31 (H-4), 3.18 (H-2)	174.0 (C-1)					
		C-4	1.31 *	21.6	4.12 (H-3)	62.6 (C-2), 67.7 (C-3)					
**10**	**4-Hydroxy-isoleucine**	C-2	3.81 (d, *J* = 5.5 Hz)	58.7	1.82 (H-3)	13.6 (C-6), 43.4 (C-3), 175.1 (C-1)	-	-	+	-	
		C-3	1.82 (m)	43.4	3.81 (H-2), 3.79 (H-4)	13.6 (C-6), 22.9 (C-5), 58.7 (C-2), 72.0 (C-4), 175.1 (C-1)					
		C-4	3.79 *	72.0	1.82 (H-3), 1.24 (H-5)	58.7 (C-2), 72.0 (C-4),					
		C-5	1.24 (d, *J* = 6.3 Hz)	22.9	3.79 (H-4)	43.4 (C-3), 72.0 (C-4)					
		C-6	0.99 (d, *J* = 7.2 Hz)	13.6	3.79 (H-4)	43.4 (C-3), 58.7 (C-2), 72.0 (C-4)					
**11**	**Asparagine**	C-2a	2.72 (dd, *J* = 9.3, 17.0 Hz)	36.2	3.84 (H-3)	53.4 (C-3), 174.7, 176.4 (C-1 and C-4)	+	+	+	+	
		C-2b	2.94 (dd, *J* = 3.6, 17.0 Hz)	36.2	3.84 (H-3)	53.4 (C-3), 174.7, 176.4 (C-1 and C-4)					
		C-3	3.84 *	53.4	2.72 (H-2a), 2.94 (H-2b)	36.2 (C-2), 174.7, 176.4 (C-1 and C-4)					
**12**	**Choline**	*N*-(CH_3_)_3_	3.21 (s)	55.4	-	55.4 (*N*-CH_3_), 69.3 (*N*-CH_2_)	+	+	+	+	
		*N*-CH_2_	3.47 *	69.3	4.00 (O-CH2)	55.4 (*N*-CH_3_), 69.3 (*N*-CH_2_), 57.3 (*O*-CH_3_)					
		*O*-CH_2_	4.00 *	57.3	3.47 (N-CH2)	-					
**13**	**Betaine**	*N*-(CH_3_)_3_	3.27 (s)	52.7	-	52.7 (*N*-CH_3_), 66.7 (C-2), 168.3 (CO)	+	+	+	+	
		N-CH_32_	3.83 (s)	66.7	-	53.7 (*N*-CH_3_), 168.3 (CO)					
**14**	**Proline**	CO	-	173.6	-	-	+	+	+	+	
		C-1	3.98 *	62.0	2.11 (H-2)	24.2 (C-3), 29.6 (C-2), 173.6 (CO)					
		C-2	2.11, 2.30 (m)	29.6	3.98 (H-1)	46.2 (C-4), 62.0 (C-1)					
		C-3	1.96 (m)	24.2	3.25, 3.37 (H-4)	46.2 (C-4), 62.0 (C-1)					
		C-4	3.25, 3.37 *	46.1	1.96 (H-3)	24.2 (C-3), 29.6 (C-2)					
**15**	**Phenylalanine**	C-3	3.33 *	38.80	-	55.9, (C-2), 131.5 (C-3′/C-5′), 138.5 (C-1′)	-	+	+	+	
		C-4′	7.33 *	129.0	-	131.5 (C-3′/C-5′)					
		C-3′/C-5′	7.33 *	131.0	7.28 (H-2′/H-6′)	138.5 (C-1′)					
		C-2′/C-6′	7.28 (d, *J* = 6.7 Hz)	129.0	7.33 (H-3′/H-5′)	38.8 (C-3), 131.5 (C-3′/C-5′)					
**16**	**Tyrosine**	C-3′/C-5′	6.76 *	112.5	7.12 (H-2′/H-6′)	-	-	+	+	+	
		C-2′/C-6′	7.12 (d, *J* = 8.5 Hz)	131.0	6.76 (H-3′/H-5′)	36.6 (C-3), 157.7 (C-4′)					
**17**	**L-Dopa**	*C-2*	3.70 *	56.6	2.86, 3.17 (H-3)	120.9 (C-2′, C-6′), 173.6 (C-1)	-	-	-	+	
		C-3a	2.86 (dd, *J* = 14.7, 9.0 Hz)	36.6	3.70 (H-2)	56.6 (C-2), 116.5 (C-5′), 120.9 (C-2′,C-6′), 127.9 (C-1′), 173.6 (C-1)					
		C-3b	3.17 (dd, *J* = 14.7, 4.2 Hz)	36.6	3.70 (H-2)	56.6 (C-2), 116.5 (C-5′), 120.9 (C-2′,C-6′), 127.9 (C-1′), 173.6 (C-1)					
		C-2′	6.73 (br. s)	120.9	6.61 (H-6′)	36.6 (C-3), 127.9 (C-1′), 145.5 (C-4′)					
		C-5′	6.75 *	116.5	6.61 (H-6′)	36.6 (C-3), 120.9 (C-2′,C-6′), 127.9 (C-1′), 145.5 (C-4′)					
		C-6′	6.61 (dd, *J* = 8.1, 2.1 Hz)	120.9	6.73 (H-2′), 6.75 (H-5′)	36.6 (C-3), 116.5 (C-5′), 145.5 (C-4′)					
**18**	**Tryptophan**	C-5	7.19 (s)	125.7	-	109.9 (C-4), 129.2 (C-11), 138.9 (C-6)	+	+	+	+	
		C-7	7.33 *	113.1	7.10 (H-8)	121.1 (C-9), 129.2 (C-11)					
		C-8	7.10 (t, *J* = 7.8 Hz)	123.0	7.33 (H-7)	119.7 (C-10), 138.9 (C-6)					
		C-9	7.05 *	121.1	7.63 (H-10)	113.1 (C-7), 129.2 (C-11)					
		C-10	7.63 (d, *J* = 7.8 Hz)	119.7	7.05 (H-9)	123.0 (C-8), 138.9 (C-6)					
**19**	**Histidine**	C-5	7.75 (s)	136.5	-	134.2 (H-4), 117.0 (H-6)	+	+	+	+	
		C-6	7.01 (s)	117.0	-	134.2 (H-4), 136.5 (H-5)					
**20**	**Cytosine**	C-5	5.70 (d, *J* = 8.7 Hz)	97.1	8.01	-	+	+	+	+	
		C-6	8.01 (d, *J* = 8.7 Hz)	143.6	5.70	156.4 (C-2), 167.0 (C-4)					
**21**	**4-Aminobutyric acid**	C-2	2.36 (t, *J* = 6.9 Hz)	35.5	1.88 (H-3)	24.9 (C-3), 41.4 (C-4), 180.2 (C-1)	+	+	+	+	
		C-3	1.88 (m)	24.9	2.36 (H-2), 2.96 (H-4)	35.2 (C-2), 41.4 (C-4), 180.2 (C-1)					
		C-4	2.96 *	41.4	1.88 (H-3)	24.9 (C-3), 35.2 (C-2)					
**22**	**Acetic acid**	CH_3_	1.92 (s)	22.8	-	174.0 (CO)	-	+	-	-	
**23**	**Fumaric acid**	C-2/C-3	6.67 (s)	136.9	-	173.2 (C-1, C-4)	+	+	+	+	
**24**	**β-Sitosterol**	C-6	5.34 *	123.0	-	-	+	+	+	+	
		C-18	0.72 (s)	14.9	-	41.4 (C-12), 44.0 (C-13), 58.9 (C-14)	+	+	+	+	
		C-19	1.02*	20.2	-	143.3 (C-5)					
		C-26/C-27	0.83 *	19.6	-	47.5 (C-24)					
**25**	**Trigonelline**	C-2	9.23 (s)	148.2	-	49.4 (*N*-CH_3_), 141.0 (C-3), 146.9 (C-4), 168.4 (CO)	+	+	+	+	
		C-4	8.91 (d, *J* = 8.1 Hz)	146.9	8.07 (H-5)	147.6 (C-6), 168.4 (CO)					
		C-5	8.07 (dd, *J* = 8.1, 6.2 Hz)	129.0	8.91 (H-4), 8.88 (H-6)	141.0 (C-3), 146.9 (C-4)					
		C-6	8.88 d (*J* = 6.2 Hz)	147.6	8.07 (H-5)	49.4 (*N*-CH_3_), 129.0 (C-5), 148.2 (C-2)					
		*N-*CH_3_	4.44 (s)	49.4	-	147.6 (C-6)					
**26–31**	**Isoflavone derivatives**	*C-3′/C-5′*	6.99 *	115.7	7.49 (H-3′/H-5′)	124–128 (C-3, C-1′), 131.8 (C-2′/C-6′), 161.7 (C-4′)	+	-	-	-	
		*C-2′/C-6′*	7.49 *	132.3	6.99 (H-2′/H-6′)	115.7 (C-3′/C-5′), 124–128 (C-3, C-1′), 161.7 (C-4′)					
**26–28**	**Genistein derivatives**	C-2	8.08 (s)	155.5	-	124–128 (C-3, C-1′), 160.9 (C-9), 182.9 (C-4)	+	-	-	-	
			8.17 (s)	156.2	-	124–128 (C-3, C-1′), 159.9 (C-9), 183.7 (C-4)					
			8.20 (s)	156.6	-	124–128 (C-3, C-1′), 159.9 (C-9), 183.7 (C-4)					
**29–31**	**Daidzein derivatives**	C-2	8.15 (s)	155.6	-	124–128 (C-3, C-1′), 159.9 (C-9), 179.3 (C-4)	+	-	-	-	
			8.23 (s)	156.2	-	124–128 (C-3, C-1′), 179.3 (C-4)					
			8.27 (s)	156.2	-	124–128 (C-3, C-1′), 159.9 (C-9), 179.3 (C-4)					
**26**	**Biochanin-A**	C-6	6.23 (d, *J* = 2.1 Hz)	101.2	6.35 (H-8)	107.2 (C-10), 167.0 (C-7)	+	-	-	-	
		C-8	6.35 (d, *J* = 2.1 Hz)	95.1	6.23 (H-6)	101.2 (C-6), 160.9 (C-9), 167.0 (C-7)					
		*O*-CH_3_	3.83 (s)	56.0	-	161.7 (C-4′)					
**27**	**Genistin**	C-6	6.52 (d, *J* = 2.1 Hz)	101.8	6.71 (H-8)	96.5 (C-8), 109.0 (C-10), 164.6 (C-5)	+	-	-	-	
		C-8	6.71 (d, *J* = 2.1 Hz)	96.5	6.52 (H-6)	101.8 (C-6), 109.0 (C-10), 159.9 (C-9), 165.3 (C-7)					
		C-1″	5.06 (d, *J* = 7.8 Hz)	102.5	3.50 (H-2″)	78.3 (C-2″), 165.3 (C-7)					
**28**	**Malonyl-genistin**	C-6	6.50 (d, *J* = 2.3 Hz)	101.8	6.72 (H-8)	96.5 (C-8), 109.0 (C-10), 164.6 (C-5)	+	-	-	-	
		C-8	6.72 (d, *J* = 2.3 Hz)	96.5	6.50 (H-6)	101.8 (C-6), 109.0 (C-10), 159.9 (C-9), 165.3 (C-7)					
		Malonyl CH_2_	3.17 (s)	42.1	-	-					
**29**	**Formononetin**	C-5	8.05 (d, *J* = 9.0 Hz)	129.0	6.94 (H-6)	160.9 (C-9), 165.3 (C-7), 179.3 (C-4)	+	-	-	-	
		C-6	6.94 (dd, *J* = 9.0, 2.3 Hz)	117.0	6.86 (H-8), 8.05 (H-5)	103.8 (C-8), 118.7 (C-10)					
		C-8	6.86 (d, *J* = 2.3 Hz)	103.8	6.94 (H-6)	117.0 (C-6), 118.7 (C-10), 160.9 (C-9), 165.3 (C-7)					
		*O*-CH_3_	3.83 (s)	56.0	-	161.7 (C-4′)					
**30**	**Daidzin**	C-5	8.14 (d, *J* = 8.7 Hz)	129.0	7.25 (H-8)	105.8 (C-8), 159.9 (C-9), 164.6 (C-7)	+	-	-	-	
		C-6	7.19 (dd, *J* = 8.7, 2.2 Hz)	117.7	8.14 (H-5)	105.8 (C-8), 164.4 (C-7)					
		C-8	7.25 (d, *J* = 2.2 Hz)	105.8	7.19 (H-6), 8.14 (H-5)	117.7 (C-6), 159.9 (C-9), 164.4 (C-7)					
		C-1″	5.10 *	102.5	3.52 (H-2″)	78.3 (C-3″), 164.6 (C-7)					
**31**	**Malonyl-daidzin**	C-6	7.22 (dd, *J* = 8.7, 2.2 Hz)	117.7	8.14 (H-5)	105.8 (C-8), 164.4 (C-7)	+	-	-	-	
		C-8	7.27 (d, *J* = 2.4 Hz)	105.8	7.22 (H-6), 8.14 (H-5)	117.7 (C-6), 159.9 (C-9), 164.4 (C-7)					
		Malonyl CH_2_	3.17 (s)	42.1	-	-					
**32**	**Cicerin**	*O*CH_2_*O*	5.98 *	103.2	-	150.3 (C-4′)	+	-	-	-	
		C-6	5.98 *	94.5	-	-					
		C-8	5.96 (br. s)	91.9	-	-					
		C-3′	6.37 *	98.5	-	150.3 (C-4′), 156.5 (C-2′)					
		C-6′	6.80 *	106.4	-	150.3 (C-4′), 156.5 (C-2′)					

C, *Cicer*; L, *Lens*; T, *Trigonella*; V, *Vicia*; (+), present; (-), absent; nd, not detected; *, overlapped.

**Table 2 molecules-26-00761-t002:** ^1^H-NMR quantification of major primary and secondary metabolites in different samples of legume sprouts methanol extracts (C, *Cicer*; L, *Lens*; T *Trigonella*; V, *Vicia*). Values are expressed as μg/mg dry powder ± S.D (*n* = 3), see experimental section. Chemical shifts used for metabolite quantification were determined in methanol-*d*_6_ and expressed as relative values to HMDS (0.94 mM final concentration) Statistical analysis is carried out by one-way analysis of variance (ANOVA) where unshared letters between groups are the significance value at *p* ≤ 0.05.

ID	Compound	Amount µg/mg Dry Matter			
		C	L	T	V
**1**	**ω-6 Fatty acid**	51.19 ± 4.58 ^a^	41.32 ± 4.26 ^bc^	38.16 ± 1.78 ^c^	47.77 ± 4.12 ^ab^
**2**	**ω-3 Fatty acid**	20.12 ± 1.76 ^a^	11.96 ± 0.85 ^b^	21.69 ± 0.49 ^a^	13.00 ± 1.30 ^b^
**3**	**Sucrose**	239.82 ± 6.98 ^a^	144.67 ± 5.87 ^c^	178.74 ± 3.39 ^b^	172.96 ± 7.80 ^b^
**4**	**Fructose**	148.39 ± 3.67 ^a^	82.91 ± 2.50 ^d^	103.72 ± 1.55 ^b^	95.60 ± 3.81 ^c^
**5**	**α-Glucose**	36.89 ± 5.11 ^c^	73.43 ± 5.28 ^b^	94.92 ± 1.76 ^a^	71.06 ± 7.58 ^b^
**6**	**β-Glucose**	43.15 ± 3.33 ^c^	76.31 ± 5.11 ^b^	89.00 ± 7.70 ^a^	81.72 ± 4.15 ^ab^
**7**	**Alanine**	31.46 ± 1.88 ^b^	23.51 ± 2.51 ^c^	44.59 ± 1.88 ^a^	25.01 ± 1.19 ^c^
**8**	**Valine**	12.61 ± 0.63 ^b^	14.59 ± 0.95 ^a^	12.57 ± 0.75 ^b^	10.26 ± 0.83 ^c^
**10**	**4-Hydroxyisoleucine**	0.0 ± 0.0 ^b^	0.0 ± 0.0 ^b^	51.13 ± 3.53 ^a^	0.0 ± 0.0 ^b^
**11**	**Asparagine**	61.05 ± 4.51 ^b^	73.46 ± 8.09 ^b^	93.43 ± 4.29 ^a^	72.71 ± 9.68 ^b^
**12**	**Choline**	19.06 ± 0.54 ^a^	16.91 ± 1.09 ^b^	9.06 ± 0.20 ^c^	9.94 ± 0.95 ^c^
**13**	**Betaine**	12.98 ± 0.62 ^b^	10.06 ± 0.87 ^bc^	5.06 ± 1.10 ^c^	109.16 ± 5.49 ^a^
**15**	**Phenylalanine**	0.0 ± 0.0 ^b^	8.61 ± 0.62 ^a^	8.69 ± 0.94 ^a^	9.07 ± 1.45 ^a^
**16**	**Tyrosine**	0.0 ± 0.0 ^c^	8.59 ± 0.53 ^b^	8.93 ± 0.68 ^b^	15.57 ± 2.69 ^a^
**17**	**L-Dopa**	0.0 ± 0.0 ^b^	0.0 ± 0.0 ^b^	0.0 ± 0.0 ^b^	112.40 ± 13.16 ^a^
**18**	**Tryptophan**	24.16 ± 5.02 ^a^	22.82 ± 3.70 ^a^	22.05 ± 2.32 ^a^	10.36 ± 2.99 ^b^
**19**	**Histidine**	4.23 ± 0.25 ^c^	11.22 ± 1.77 ^a^	7.43 ± 1.91 ^b^	11.07 ± 1.77 ^a^
**20**	**Cytosine**	9.39 ± 1.95 ^a^	6.16 ± 1.30 ^b^	5.53 ± 0.72 ^b^	7.30 ± 1.45 ^ab^
**22**	**Acetic acid**	0.0 ± 0.0 ^b^	10.51 ± 0.46 ^a^	0.0 ± 0.0 ^b^	0.0 ± 0.0 ^b^
**23**	**Fumaric acid**	2.18 ± 0.19 ^c^	2.51 ± 0.25 ^bc^	3.11 ± 0.16 ^a^	2.84 ± 0.34 ^ab^
**24**	**β-Sitosterol**	8.95 ± 0.67 ^b^	10.12 ± 0.78 ^ab^	8.56 ± 1.03 ^b^	10.77 ± 0.90 ^a^
**25**	**Trigonelline**	18.03 ± 0.97 ^b^	8.11 ± 1.02 ^d^	24.73 ± 1.02 ^a^	11.59 ± 1.34 ^c^
**26**	**Biochanin A**	32.04 ± 2.12 ^a^	0.0 ± 0.0 ^b^	0.0 ± 0.0 ^b^	0.0 ± 0.0 ^b^
**27**	**Genistin**	43.86 ± 4.87 ^a^	0.0 ± 0.0 ^b^	0.0 ± 0.0 ^b^	0.0 ± 0.0 ^b^
**28**	**Malonyl-genistin**	78.88 ± 1.46 ^a^	0.0 ± 0.0 ^b^	0.0 ± 0.0 ^b^	0.0 ± 0.0 ^b^
**29**	**Formononetin**	35.52 ± 2.00 ^a^	0.0 ± 0.0 ^b^	0.0 ± 0.0 ^b^	0.0 ± 0.0 ^b^
**30**	**Daidzin**	49.27 ± 3.10 ^a^	0.0 ± 0.0 ^b^	0.0 ± 0.0 ^b^	0.0 ± 0.0 ^b^
**31**	**Malonyl-daidzin**	80.22 ± 3.56 ^a^	0.0 ± 0.0 ^b^	0.0 ± 0.0 ^b^	0.0 ± 0.0 ^b^
**32**	**Cicerin**	33.19 ± 2.84 ^a^	0.0 ± 0.0 ^b^	0.0 ± 0.0 ^b^	0.0 ± 0.0 ^b^

C, *Cicer*; L, *Lens*; T, *Trigonella*; V, *Vicia*; (-), not detected.

## Data Availability

Data is available from authors upon request.

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
