# Peer review of "Nuclear Magnetic Resonance Metabolomics Approach for the Analysis of Major Legume Sprouts Coupled to Chemometrics"

_molecules, 2021, doi:10.3390/molecules26030761_

Round 1
Reviewer 1 Report
The study presents NMR identification and quantitation of metabolites characterizing the profiles of 4 different legume sprouts. A thorough description of the spectral profiles and the specific metabolites for each sprout is present, providing informative content for the readers and novelty to the study. For the statistical analysis part, the low number of samples and the choice of methods, limit the robustness of the results, which need to be further validated.
Here below some suggestions to improve the English language and the statistical validity of the study.
Abstract
Line 21: add “a” before source.
Line 24: “metabolites” not “Metabolites”.
Line 27: remove “for” before metabolome.
Line 31: rephrase “suggestive for its role”.
Line 31: change “It was most enriched” with “It was found at greater concentrations in”
Line 32: change “afforded” with “provided”.
Line 33: change “emanated” with “originated”.
Line 35: “;” instead of “,” before “NMR”.
Introduction
Line 45: Insert “,” after importance.
Line 48: “limits”.
Line 49-50: Needs to be specific about the harmful action or actions (?) of alkaloids. A reference is recommended, too.
Line 50-52: Rephrase. Perhaps “Germination is a food processing technique, in addition to cooking, autoclaving, and soaking, improving the nutritional quality of legumes by decreasing the content of antinutritional factors”.
Line 79: substitute “moreover” with “even more” or “more so”.
Results and Discussion
Figure 1a and 1b: T spectrum seem to suffer a little bit of phasing problems, did you try adjusting it?
Line 137: rephrase to “yet were not observed in Cicer samples. These amino acids were phenylalanine and tyrosine.”
Line 143: rephrase to “and can play a role in anxiety and depression relief”.
Line 156: “, such as phenolic compounds,”.
Line 156: start a new sentence with “This would thus be reflected”.
Line 158 and 163 start the same way. Rephrase one of them at least. For example it could be “The 1H-NMR spectra showed low intensity signals attributed to cytosine”. Most of the results are also presented in the past tense, so check for consistency.
Line 177: “responsible for”.
Line 177: “as well as food”.
Line 234: “levels were”.
Line 241: “,” before “amounting”.
Line 275: “were found”.
Line 309: substitute “analyses” with “models”.
Line 324: “revealed a clear discrimination”.
Line 333: “between the two sample groups”. Otherwise it seems there are only 2 samples.
PLS-DA like that, in a small sample set, with no CV, can be considered a little overfitting. Did the authors try a simple PLS prediction using maybe some of the metabolite markers? Univariate analysis to check for statistical difference of the metabolite markers is also recommended, in spite of the very small sample size. Box plot showing the differences in specific markers would be more helpful to the reader than loading plots or S-plots.
Materials and methods
Line 344: remove “;” after fava.
Line 361: “developed by”.
Line 368: It is best to report a volume of the NMR sample. Sampled supernatant needs to be homogeneous.
Was a calibration curve for quantitation checked for specific metabolites? If so, this could be a useful information/figure to include in Supplementary, due to the novelty of the study.
Conclusions
Mention the need for further validation of the biomarkers with larger sample sets.
Author Response
see attached file please

Reviewer 2 Report
Nuclear Magnetic Resonance Metabolomics Approach for the Analysis of Major Legume Sprouts Coupled to Chemometrics
In this study, authors have provided the first NMR based metabolite fingerprinting of 4 major legume sprouts; Chickpea (Cicer arietinum), fenugreek (Trigonella foenum-graecum), fava beans (Vicia faba), and lentil (Lens esculenta). A total of 32 compounds from various metabolite classes were identified and quantified. PCA and OPLS-DA were used for assessing the variations and determining main markers to be utilized in quality control assessment. Trigonelline which was detected in all legume sprouts and 4-hydroxy-isoleucine were found more enriched in Trigonella. Higher amount of isoflavonoids and sucrose were detected in Cicer sprout while Vicia was characterized by the exclusive presence of L-Dopa and Lens revealed higher abundance of acetate. The aromatic region data provided a better classification model than the full range NMR as legume sprout variations mainly contributed by secondary metabolites.
The work presented in the manuscript is an extension from the recently published manuscript from their team titled “Mass spectrometry-based metabolites profiling of nutrients and anti-nutrients in major legume sprouts” (Farag MA, El-Din MGS, Selim MA, Owis AI, Abouzid SF. Mass spectrometry-based metabolites profiling of nutrients and anti-nutrients in major legume sprouts. Food Bioscience, 2020, 100800). The current study provides information on structure elucidation and quantification of influential metabolites.
Overall comments:
The manuscript contains noteworthy information on the metabolites of the studied legume sprouts. The characterization of metabolites was supported with comprehensive 2D NMR experiments. Overall, the experimental design, methods and the results are well described. However, multivariate data analysis can be vastly improved, as suggested below. There are some additional areas related to methodological and other types of data analysis will need to be addressed, in order to improve the quality and clarity of the information presented. A major drawback of the study is that no generalization can be made as only one variety was used for each genus. Hence, the claims have to be made narrower than as stated in this version of the manuscript.
Specific comments:
- Please provide some background information on the four selected major legume sprouts, such as varieties and commercial source.
- Line 96. The standardization of extracts is important to ensure their quality especially if the end goals are to utilize legume sprout as raw materials in nutraceuticals application. Why the authors choose to use methanol as solvent of extraction, rather than opting for any biphasic or different ratios of methanol as extraction solvent? Please explain this.
- Line 99. The authors have performed comprehensive 2D NMR experiments allowing the structure elucidation to be accomplished. Why the 2D J-resolved experiment was not in the list? Please comment on this.
- Table 1. It is interesting that the isoflavonoids identified as compounds 26 to 32 were only detected in the Cicer samples from this one variety characterized in the study. Please highlight the caution that given the single variety used in this study, it cannot be concluded that whether this isoflavonoids group is specific to genus Cicer or not.
- Line 228. Was the quantification based on absolute or relative manner?
- Table 2. Please include statistical significance of the computed data.
- Line 306. The OPLSDA loading plots (S-plot) represents the chemical shifts of the characterized compounds. The compounds that positions at the far end of the S-plot usually are considered as the prominent metabolites contributing towards the observed clustering. Please explain or include the basis for emphasizing the one/two compounds selected from the loading plots shown in Figures S14B, S15B, S16B, S17B, S18B and S19B?
- The influential metabolites have been determined using OPLS-DA loading plots. The authors are encouraged to include information on the VIP values of these metabolites. This can further validate the assignment of markers differentiating legume sprouts.
- Line 340. In order to improve multivariate statistical information, authors can include analyses that reveal identity of the influential metabolites/relationship between metabolite and legume sprouts. These can be done using but not restricted to tools such as visualization of metabolite abundance using heatmap.
- Line 342. The research aim was to offer information on standardized extracts of the four major legume sprouts. Parameters such as growth stage, harvesting time are known to affect plant metabolome. The conditions and details related to sample collection and processing should be included so that other researchers can replicate the experimental design.
- Line 357. How much was the final concentration of the HMDS used in this study?
- Line 358. The buffered NMR solvents are regularly being used to normalize pH across samples. This helps to minimize chemical shift variability and peak alignment. However, no details on usage of buffer was included in the methodology section. Please include how the pH variation was regulated in this study.
- Line 358. Please state the name of other chemical standards.
- Line 416. Where are the results of the Distance to the model (DModX) tests?
Author Response
see attached file please

Reviewer 3 Report
The manuscript with the title “Nuclear Magnetic Resonance Metabolomics Approach for the Analysis of Major Legume Sprouts Coupled to Chemometrics” is focusing on classification of the sprouts sample. The authors write about the application of metabolomics providing important metabolites for the discrimination.
Major Points:
- In the introduction it becomes not clear why there is now a focus on NMR metabolomics? Previous studies should be added to clarify the importance of NMR metabolomics.
- Figure 3. Sugars had a negative effect and sucrose had a positive effect on PC1. This should be discussed and explained.
- OPLS-DA results provided main markers for the discrimination. So, the figure results should be added in the main manuscript.
Author Response
see attached file please

Round 2
Reviewer 2 Report
Comparative Nuclear Magnetic Resonance Metabolomics Approach for the Analysis of Major Legume Sprouts Coupled to Chemometrics
The authors have sufficiently improved their paper by addressing the reviewer’s comments and suggestions. Thorough responses have been provided to the queries from the previous review. However, some responses need further clarification and below are some additional comments for the authors to consider.
1. Response: As previously mentioned, an additional statement was at the end of the conclusion in the new version highlighting the need for further testing of other varieties within each legume seed; see our response above.
Comment: As the caution highlighted as direct interpretation of the results, the authors also need to include it in discussion.
2. Response: 3.9. Statistical analysis. NMR quantification data were analysed using the Co-Stat computer program (version 8). Data are expressed as mean ± SD of the groups. Differences between sample groups were compared by one-way analysis of variance (ANOVA) and were considered statistically significant when p ≤ 0
Comment: The standard norm for statistical significance is at p < 0.05
3. Response: We provided heatmaps showing metabolites abundance from full and aromatic regions in supplementary figures (Figure S15)
Comment: For panel A of Figure S15, the labels of the chemical shifts are overcrowded such that the values cannot be seen. Please revise the annotation for better clarity.
Author Response
see attached letter
